# Fracture Resistance of Partial Indirect Restorations Made with CAD/CAM Technology. A Systematic Review and Meta-Analysis

**DOI:** 10.3390/jcm8111932

**Published:** 2019-11-09

**Authors:** Amaia Amesti-Garaizabal, Rubén Agustín-Panadero, Blanca Verdejo-Solá, Antonio Fons-Font, Lucía Fernández-Estevan, Jose Montiel-Company, María Fernanda Solá-Ruíz

**Affiliations:** Stomatology Department School of Medicine and Dentistry, University of Valencia, 46010 Valencia, Spain; amaiaamesti@gmail.com (A.A.-G.); blanca.verdejo@gmail.com (B.V.-S.); antonio.fons@uv.es (A.F.-F.); lucia.fernandez-estevan@uv.es (L.F.-E.); jose.maria.montiel@uv.es (J.M.-C.); m.fernanda.sola@uv.es (M.F.S.-R.)

**Keywords:** partial indirect restorations, inlays, onlays, overlays, ceramics, hybrid material, composite, fracture resistance

## Abstract

Background: The aim of this systematic review and meta-analysis was to determine the fracture resistance and survival rate of partial indirect restorations inlays, onlays, and overlays fabricated using computer-aided design and computer-aided manufacturing (CAD-CAM) technology from ceramics, composite resin, resin nanoceramic, or hybrid ceramic and to analyze the influence of proximal box elevation on fracture resistance. Materials and methods: This systematic review was based on guidelines proposed by the preferred reporting items for systematic reviews and meta-analyses (PRISMA). An electronic search was conducted in databases US National Library of Medicine National Institutes of Health (PubMed), Scopus, Web of Science (WOS), and Embase. In vitro trials published during the last 10 years were included in the review. Results: Applying inclusion criteria based on the review’s population, intervention, comparison, outcome (PICO) question, 13 articles were selected. Meta-analysis by restoration type estimated the fracture resistance of inlays to be 1923.45 Newtons (N); of onlays 1644 N and of overlays 1383.6 N. Meta-analysis by restoration material obtained an estimated fracture resistance for ceramic of 1529.5 N, for composite resin of 1600 Ne, for resin nanoceramic 2478.7 N, and hybrid ceramic 2108 N. Conclusions: Resin nanoceramic inlays present significantly higher fracture resistance values. Proximal box elevation does not exert any influence on the fracture resistance of indirect restorations.

## 1. Introduction

Various treatment options are available for performing dental restorations in the posterior region. The longevity of dental restorations in this region depends on multiple factors, including the properties of the restoration material, the state of the supporting teeth, the patients’ habits, and the adopted clinical protocols [1].

High patient demand for esthetic outcomes conditions which materials are selected for fabricating restorations [2,3]. Of course, the choice of material cannot be guided by esthetics alone but must depend on its clinical behavior [4,5]. Thanks to the development of new materials, dental reduction can be minimized and the conservation of dental tissue maximized, which will contribute to the longevity of the restoration [2,4,6].

When the coronal structure of a posterior tooth is lost, various treatment options are available, depending on the degree of destruction: Direct restoration with composite, partial indirect restoration classified as inlays (without covering the cusps), onlays (covering at least one cusp), and overlays (covering all cusps), or full coverage restoration (crowns) [7].

Indirect partial restorations allow conservation of the remaining dental structure and strengthen a tooth compromised by caries or fracture [2,8]. Indirect techniques involve manufacturing the restoration away from the oral cavity in a dental laboratory, which avoids some of the difficulties of direct techniques, such as polymerization contraction and marginal adaptation [6]. However, indirect composite resin or ceramic partial reconstructions require more extensive tooth preparation. Selecting the treatment technique correctly will also contribute to conserving the maximum amount of healthy dental tissue [7].

Fracture is considered the most common reason for replacing fixed partial prostheses. For this reason, when considering the long term viability of a dental material for fabricating restorations, it is essential to test its fracture resistance. 

During the last 35 years, computer-aided design and computer-aided manufacturing (CAD-CAM) technology has played an increasing role in dentistry, allowing restoration design and fabrication by mechanized, computer-assisted techniques [9,10]. As these systems have gained in popularity, new materials have been introduced adapted to milling restorations using CAD-CAM processing [10]. 

In the context of these evolving technologies and the numerous materials that can be used, including ceramic (conventional feldspathic ceramic, leucite-reinforced ceramic, lithium disilicate ceramic), composite resin, and hybrid materials (resin nanoceramic and hybrid ceramic), this systematic review set out to determine whether or not CAD-CAM systems, and the materials they employ to fabricate indirect restorations, are viable in the medium and long term by investigating their fracture resistance.

Researchers, as well as manufacturers, seek an ideal material for fabricating partial coverage restorations, such as inlays, onlays, and overlays. In recent decades, the numbers of ceramic or ceramic-like materials available have increased dramatically, often making it difficult to define their different uses and establish clear selection criteria and adequate indications. In 2015, Gracis et al. published a classification of dental ceramics and ceramic-like materials, highlighting ceramics with vitreous matrix or glass-ceramics; this category includes conventional feldspathic ceramics, synthetic ceramics (leucite-reinforced and/or lithium disilicate), and hybrid materials or ceramics with resin matrices, e.g., resin nanoceramic and resin matrix glass ceramic (hybrid ceramic) [11].

Ceramics have been the subject of extensive scientific research, which has vouched for their use in this type of restoration. They offer adequate fracture resistance (160–450 MPa), good survival rates, a high elasticity modulus, and they do not suffer much abrasion or wear. Composite, which has seen improvements in its composition, accompanied by the introduction of CAD/CAM processing, has been presented as an alternative to ceramics that offers certain advantages, such as its resilience and repairability, although its polishing capacity, resistance to wear, and fracture resistance are poorer. It is claimed that hybrid materials, resin nanoceramics, and resin matrix glass ceramics offer the advantages of both ceramics and composite, with an elasticity modulus similar to the natural tooth’s dentine and characteristics that make these materials (like composite) easy to adjust, repair, or modify [12].

Some researchers in dental preparation propose proximal box elevation, because the technique provides increased fracture resistance, but there is great disparity in the results obtained with the technique [3,13,14,15].

The aim of this systematic review and meta-analysis was to determine the fracture resistance and survival rates of indirect restorations—inlays, onlays, and overlays—fabricated using CAD-CAM technology from ceramics, composite resin, or hybrid material and to analyze the influence of proximal box elevation on fracture resistance.

## 2. Materials and Methods

This systematic review was based on guidelines proposed by Preferred Reporting Items for Systematic Reviews and Meta-Analyses (PRISMA) An electronic search was conducted in four databases: US National Library of Medicine National Institutes of Health (PubMed), Scopus, Web of Science (WOS), and Embase. The search strategy used a combination of key terms: “inlay”, “onlay”, “overlay”, “CAD-CAM materials”, “dental porcelain”, “dental ceramic”, “composite resin”, “hybrid CAD-CAM material”, “polymer infiltrated”, “survival rate”, “fracture resistance”. They were combined applying Boolean operators “AND, OR”. The search excluded articles involving implants or full coverage crowns by applying the Boolean operator “NOT” followed by “implant” and “crown”. Each of the search terms used, and how they were combined, varied depending on the database searched.

The review’s population, intervention, comparison, outcome (PICO) items defined the search strategy: Population: Inlays, onlays, or overlays fabricated using CAD-CAM technology;Intervention: Different materials used to fabricate the indirect restorations by means of CAD-CAM technology;Comparison: not applicable;Outcome: Survival rates, fracture resistance, and study type.

The articles identified were assessed considering title, abstract, and full text, applying the following inclusion criteria: Articles published during the last 10 years; in vitro trials of survival and fracture resistance of indirect partial restoration of human posterior teeth, without language restrictions. The following exclusion criteria were applied: In vivo clinical trials, trials not assaying fracture resistance of indirect partial restoration of posterior teeth or which did not specify the material used to fabricate indirect restorations, and studies whose topic was not relevant to the aims of the present systematic review.

The following variables were extracted from in vitro articles: Authors, journal, year of publication, sample size, material employed, indirect restoration type (inlay, onlay, overlay), follow-up period, different methods of aging, to which samples were subjected, and maximum load recorded, in Newtons (N), that produced fracture of the specimens.

Quantitative analysis was performed by means of meta-analysis. Articles were combined in a random effects model. Heterogeneity was evaluated using the Q test and *p*-value (considering heterogeneity present when *p* < 0.1) and using the I2 test (25–50% slight, 50–75% moderate, and >75% high heterogeneity). Additional analysis was performed according to the subgroup (materials or restoration type), as well as random effects regression for meta-analysis, using the maximum likelihood method. To assess publication bias, the fail-safe number was calculated (number of missing studies that should be added to make the combined effect size statistically insignificant), as well as Egger´s regression intercept (considering bias to exist when a 95% CI did not include 0). Effect size was estimated with the trim and fill method to evaluate whether the effect would change by imputing new studies to create funnel plot symmetry. 

## 3. Results

The initial electronic search identified 61 articles, of which 21 were duplicates. The titles and abstracts of the 40 remaining articles were read, discarding a further 11 works for the following reasons: Three reviewed the evolution of CAD-CAM materials, four were unrelated to indirect partial restoration, and four were unavailable via the University Network, despite having attempted to contact the authors to obtain the full text.

The full texts of the 29 remaining articles were read applying the inclusion and exclusion criteria, and a further 18 works were discarded for the following reasons: Five were in vivo studies, eight did not test the fracture resistance of indirect partial restorations on posterior teeth but focused on other risk factors, such as marginal fit or the materials’ hardness, two failed to specify the material employed to fabricate restorations, and three investigated the thickness of the restoration material in relation to fracture resistance.

Finally, a total of 13 in vitro trial articles were included in the systematic review, adding another two studies identified from the reference sections of the articles selected. The study selection procedure is illustrated in detail (Figure 1).

In vitro studies provided results in Newtons (N) of the maximum load at which fracture of the restoration material was produced. They also determined whether or not proximal box elevation influenced fracture resistance.

The samples sizes were ranging from 30 to 160 restorations. Eight articles investigated inlays, four onlays, and the other two overlays. Most of the samples were subjected to aging to simulate conditions in the oral medium as realistically as possible, with one exception (the trial conducted by Liu 2014 [3], which did not apply any aging method). Results were expressed in Newtons (N) in most of the articles, with the exception of Magne (2009) [15], Batalha-Silva (2013) [16], and Soares (2018) [17], who reported the percentage of samples surviving a specific load. 

The first analysis was carried out, focusing on studies comparing the fracture resistance of restorations in which a marginal box elevation has previously been made, with those in which the box has not been raised. A total of three articles have been used for this (Figure 2). The Q test was 12.94, with a *p*-value of 0.012 and a I2 value of 69.1%. When presenting a moderate heterogeneity, the random effects model is studied, where the average is −75.653 N, with the favorable data for the group without elevation of the marginal case. But these data are not relevant because they do not show statistical significance (lower limit −413,089 N, upper limit 261,782 N, and *p*-value of 0.660).

Meta-analysis of the data reported in 10 studies (Figure 3) comprises the fracture resistance of inlays, onlays, and overlays made from composite, ceramic, or resin nanoceramic. Estimating the effect size, the mean fracture resistance was 1729.9 N with a 95% CI between 1443.6 and 2016.2. The articles showed high heterogeneity (Q test = 267.1; *p* < 0.01; I^2^ = 94.4%). 

To determine heterogeneity, data was analyzed by subgroups—restoration type and material—combining data in random effects models. 

Meta-analysis by restoration material (Figure 4) obtained an estimated fracture resistance for ceramic of 1529.5 N (95% CI between 1110.2 and 1948.7), for composite of 1600 N (95% CI between 952.8 and 2247.2), and for resin nanoceramic of 2478.7 N (95% CI between 2401.1 and 2556.3). The ceramic subgroup showed high heterogeneity (Q test = 74.8; *p* < 0.01; I^2^ = 87.9%), the composite subgroup showed moderate heterogeneity (Q test = 3.40; *p* = 0.07; I^2^ = 70.5%), while the resin nanoceramic subgroup did not show heterogeneity (Q test = 1.99; *p* = 0.58; I^2^ = 0%).

Meta-analysis by restoration type (Figure 5) estimated the fracture resistance of inlays to be 1923.45 N (95% CI between 1594.7 and 2252.2), of onlays to be 1644 N (95% CI between 1166.2 and 2121.9), and of overlays to be 1383.6 N (95% CI between 1069.7 and 1697.5). The inlay subgroup showed high heterogeneity (Q test = 121.2; *p* < 0.01; I^2^ = 94.2%), the onlay subgroup did not present heterogeneity (Q test = 4.12; *p* = 0.39; I^2^ = 2.9%), and the overlay subgroup showed moderate heterogeneity (Q test = 7.22; *p* = 0.03; I^2^ = 72.3%).

Meta-regression analysis was performed by means of a random effects model using the method of maximum likelihood with restoration type and restoration material as covariables. The model obtained statistical significance (Q test *p* = 0.015; R^2^ = 0.65). Data are shown in Table 1. 

The model did not identify differences between types of restoration (Figure 6), but different restoration materials (Figure 7) did show a significant difference in fracture resistance of 670.9 N (95% CI between 148.5 and 1193.4) for resin nanoceramic in comparison with ceramic, which acted as a reference. The fail-safe number is 188957, indicating that 188957 “null” studies would be needed for the combined 2-tailed *p*-value to exceed 0.05. The Egger’s intercept test obtained −7.498; 95% confidence interval (−18.72; 3.73) with *t* = 1.432 and df = 14. The *p* value was 0.170. 

The random effects model obtained a point estimate of effect for the combined studies of 1729.9 and 95% confidence interval (1443.6; 2016.2). These values were unchanged using the trim and fill method.

## 4. Discussion

In the present systematic review, the combined evidence drawn from in vitro trials, regarding the fracture resistance of restorations fabricated using CAD-CAM technology, was not homogenous. The selected process was conducted meticulously to make the total as uniform as possible and allow comparison of the results, but nevertheless suffered major limitations, in particular the variations in follow-up time. 

Despite the limitations of in vitro trials, they are necessary for determining the properties of different restoration materials before they are applied in vivo with satisfactory results. 

In vitro trials used healthy teeth (molars or premolars), extracted mainly for orthodontic reasons, including vital teeth and teeth treated endodontically. The materials compared were conventional feldspathic ceramic, leucite-reinforced ceramic, lithium disilicate ceramic, composite, and hybrid materials, used to fabricate inlays, onlays, and overlays.

To bond the restorations, all the studies employed resin-based cements (self-curing, light-curing, or dual-curing), and, when cemented, most specimens underwent some aging process (mechanical or thermocycling). 

### 4.1. Inlays

Among the articles assaying leucite-reinforced ceramic, Keshvad et al. compared the fracture resistance of 15 molars restored with inlays fabricated from leucite-reinforced ceramic IPS Empress CAD (Ivoclar Vivadent, Schaan, Liechtenstein) ceramic. After subjecting the specimens to aging (storing samples in distilled water at 37 °C for 24 h followed by thermocycling), a universal test machine (Z020 Zwick/Roell) obtained a maximum load at the moment of fracture of 1050 ± 763 N [18].

Liu et al. carried out a trial using leucite-reinforced ceramic to restore 16 molars (divided into two groups of eight specimens, with and without proximal box elevation), with compression loading until fracture. The maximum load obtained was 1799.78 ± 338.88 N, being higher in the group with proximal box elevation (2004.89 ± 183.59 N; *n* = 8). This is a very different result from the trial by Keshvad; a difference that could be due to the fact that the samples were not aged in the latter work [3].

The article by Sener-Yamaner et al. had a sample of 20 molars restored with lithium disilicate ceramic inlays (IPS e.max CAD, Ivoclar Vivadent, Schaan, Liechtenstein), stored in distilled water at 37 °C for a week and then evaluated using a universal test machine (Shimadzu AG-IS, Kyoto, Japan). The mean fracture resistance was 2007 ± 29.5 N and 2594 ± 35.52 N in the control group [19].

Yoon et al. also assayed lithium disilicate ceramic inlays, comparing the fracture resistance of 20 inlays restoring premolars. After aging the samples (stored in distilled water at 37 °C for 24 h followed by thermocycling), a universal test machine (Instron 3366, Instron, Norwood, MA, USA) obtained a mean maximum load of 661.85 ± 302.95 N [20].

Three trials published in 2016 tested the fracture resistance of inlays fabricated from resin nanoceramic. Tekçe restored a sample of 10 endodontically-treated molars with Lava Ultimate (3 M ESPE, St Paul, MN, USA) inlays, afterwards stored in distilled water until they were evaluated using a universal test machine (Instron 6022, Instron Corp., Norwood, MA, USA), obtaining a maximum load of 2340.7 ± 366.52 N, with significant difference in comparison with a control group 2815.8 ± 222.67 N [21].

The same author conducted another similar trial, but with vital molars restored with inlays made from hybrid ceramic (Cerasmart, GC America, Alsip, IL, USA). The sample size was 10 specimens per group, and all were stored in distilled water for 24 h at 37 °C until evaluated with a universal test machine (Instron 6022, Instron, MA, USA). The mean fracture resistance was 2108.4 ± 464.56 N, obtaining higher fracture resistance in a group with juxta-gingival cavities [13].

Soares et al. made a study of the same material, comparing the fracture resistance of 15 molars restored with inlays, stored for a week in distilled water at 37 °C before testing with a universal testing machine Acumen 3 (MTS Systems, Eden Prairie, MN, USA). The specimens obtained a mean survival rate of 93% when subjected to a load of 1675 N [17].

Sener-Yamaner assayed inlays made from Lava Ultimate (3 M ESPE, St Paul, MN, USA), with a sample of 20 molars, aged and loaded in a universal test machine (Shimadzu AG-IS), obtaining a mean load at the point of fracture of 2486 ± 40.06 N, which was slightly lower than the control group (2594 ± 35.52 N) [19].

As for inlays fabricated from composite Paradigm MZ100 (3M-ESPE, St. Paul, MN, USA), two trials were located: One by Batalha-Silva et al. with 17 restored and aged molars, obtaining a survival rate of 100% when specimens were subjected to a load of 1400 N [16]. The other by Liu (2014), who evaluated a sample of 16 non-aged molars divided into two groups, subjected to an axial force of up to 2200 N or until fracture, obtaining a mean fracture resistance of 2030.41 ± 359.78 N, with slightly higher values when the proximal box elevation was performed (2057 ± 293.88 N) [3].

In light of the above, it can be seen that inlay restorations obtain higher fracture resistance when fabricated from resin nanoceramic, followed by lithium disilicate ceramic, and composite, leucite-reinforced ceramic inlays obtained the lowest fracture resistance.

These observations were confirmed in meta-analysis, which found statistically significant differences between ceramic, composite, and resin nanoceramic, the latter obtaining the best results. 

### 4.2. Onlays

For onlays fabricated from conventional ceramic, Ilgenstein et al. obtained a fracture resistance of 1373.6 ± 661.1 N, with a sample of 12 specimens (*n*); this being higher than restorations with proximal/marginal box elevation [14].

In the same trial, Ilgenstein also assayed the fracture resistance of onlays fabricated from nanoceramic resin. In this case the sample of *n* = 12 underwent aging (thermocycling and simulated mastication), obtaining values of 1828.8 ± 596.45 N, although specimens with marginal box elevation produced lower values [14].

Yildiz et al. conducted a trial of 20 specimens of lithium disilicate ceramic onlays, comparing a control group with two groups of 10 specimens cemented with different materials: Self-curing and dual-curing resin. The mean fracture resistance was 1970.3 ± 457.49 N; the control group (2028.98 ± 460.55 N) and the group cemented with dual-curing resin produced higher values [22].

Vianna et al. also investigated lithium disilicate ceramic, comparing the fracture resistance of 24 molars restored with onlays, evaluated with a universal testing machine (DL2000, EMIC, São José dos Pinhais, Brazil). The mean fracture resistance was 2603.85 ± 617.1 N [23].

Vianna, in the same trial, measured the fracture resistance of 24 leucite-reinforced ceramic onlays, obtaining a mean value of 1693.1 ± 465.45 N [23].

The article by Yoon cited above also tested 40 lithium disilicate onlays. After aging the specimens (storage in distilled water at 37 °C for 24 h followed by thermocycling), the samples were assayed in a universal test machine (Instron 3366, Instron, Norwood, MA, USA), obtaining a mean fracture resistance of 863.975 ± 495.45 N [20].

Descriptively, onlays fabricated from lithium disilicate ceramic offered the highest fracture resistance, especially when restorations were cemented with dual-curing resin, followed by the resin nanoceramic onlays. Conventional feldspathic onlays produced the lowest fracture resistance.

According to meta-analysis, differences did not reach statistical significance (estimated confidence intervals for the materials coincided). Therefore, the fracture resistance of onlays is not greatly influenced by the material used to fabricate them (ceramic or resin nanoceramic). 

### 4.3. Overlays

The articles analyzed in the present systematic review included overlays fabricated from three materials: Conventional feldspathic ceramic, ceramic reinforced with leucite, and composite resin. No articles were found that tested overlays made from hybrid materials or lithium disilicate ceramic, which also fulfilled the inclusion criteria applied in the review. 

All the trials of overlays used endodontically-treated molars and were cemented with similar light-curing materials.

Magne et al. compared the survival rates of two materials used to fabricate 30 molars, stored and subjected to simulated cyclic isometric chewing (5 Hz), starting with a load of 200 N (5000 cycles), followed by stages of 400, 600, 800, 1000, 1200, and 1400 N to a maximum of 30,000 cycles each. Samples were loaded until fracture or to a maximum of 185,000 cycles. A total of 15 molars were restored with conventional ceramic overlays and 15 molars with composite overlays. The results showed a survival rate of 66% (10 specimens) for ceramic overlays subjected to a load of 1000 N de, 7% (1 specimen) subjected to 1200 N, and 0% subjected to 1400 N, the mean fracture resistance of conventional ceramic being 1147 N. However, the composite overlays obtained 100% when subjected to a load of 1200 N and 73% (11 specimens) subjected to 1400 N, pointing to higher fracture resistance for composite than ceramic. In both cases, the samples sizes were the same, as was the material used for cementing the restorations, as well as the aging process applied before loading; for this reason, the differences in fracture resistance obtained can only be due to the properties of the different materials employed to fabricate the overlays [15].

A trial conducted by Dere et al. tested a sample of eight specimens restored with leucite-reinforced ceramic overlays, comparing them with a control group of intact teeth. The control group obtained a mean fracture resistance of 3048 ± 905 N, but much lower values (2036 ± 319 N) for teeth restored with leucite-reinforced ceramic overlays [24].

Descriptively, overlays obtained different fracture-resistance results depending on the material employed, leucite-reinforced ceramic obtaining the highest values, followed by composite, which showed a strength considered medium [24]. Overlays fabricated from conventional feldspathic ceramic were found to be more fragile [15].

However, the results of meta-analysis did not identify statistically significant differences (estimated intervals for ceramic and composite coincided).

## 5. Conclusions

Within the limitations of the present systematic review and meta-analysis, the following conclusions may be drawn: Maximum fracture resistance for inlays was obtained by restorations fabricated from resin nanoceramic; for onlays, maximum fracture resistance was obtained by lithium disilicate ceramic; for overlays, the highest fracture resistance values were obtained by leucite-reinforced ceramic. 

No consensus exists as to the most indicated choice of material for fabricating indirect restorations, although inlays made from resin nanoceramic have been shown to offer significantly higher fracture resistance. There is no influence between the proximal box elevation and the fracture resistance of teeth restored with indirect restorations.

## Figures and Tables

**Figure 1 jcm-08-01932-f001:**
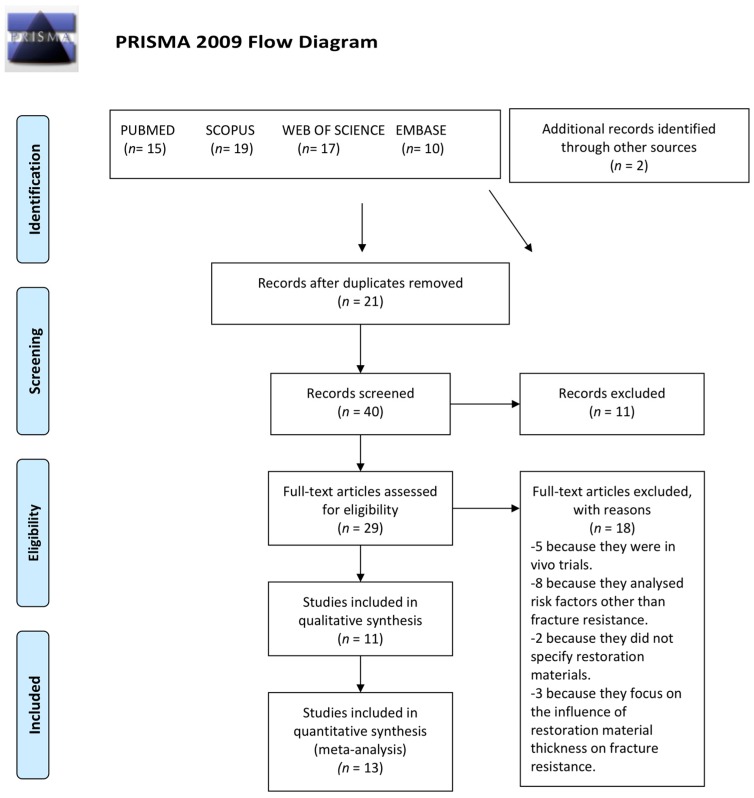
Flow chart of study selection procedure.

**Figure 2 jcm-08-01932-f002:**
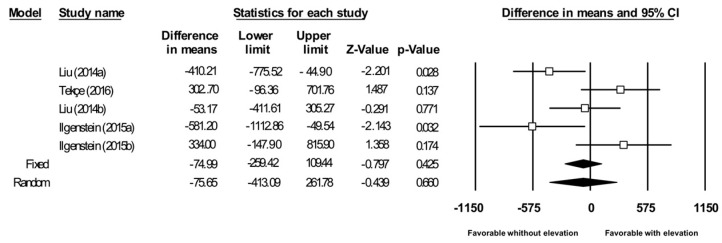
Fracture resistance (Newtons (N)) in indirect restorations with/without elevation of the marginal box. The size of the square represents the weight of each study and the diamond represents the combined results of the meta-analysis. CI is the 95% confidence interval of the mean between a lower and an upper limit.

**Figure 3 jcm-08-01932-f003:**
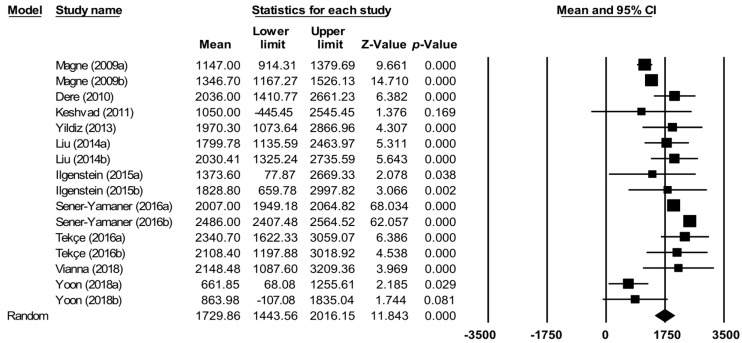
Forest plot of overall meta-analysis of fracture resistance (in Newtons) of restorations (inlays, onlays, and overlays) fabricated from different materials. The size of the square represents the weight of each study and the diamond represents the combined results of the meta-analysis. CI is the 95% confidence interval of the mean between a lower and an upper limit.

**Figure 4 jcm-08-01932-f004:**
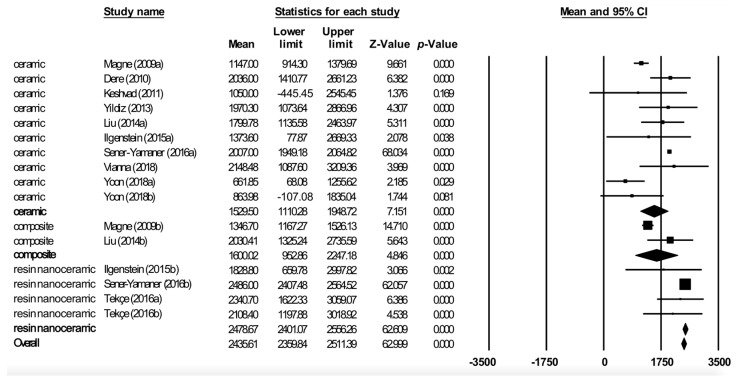
Forest plot of meta-analysis of materials subgroups (ceramic, composite, and resin nanoceramic). The size of the square represents the weight of each study and the diamond represents the combined results of the meta-analysis. CI is the 95% confidence interval of the mean between a lower and an upper limit.

**Figure 5 jcm-08-01932-f005:**
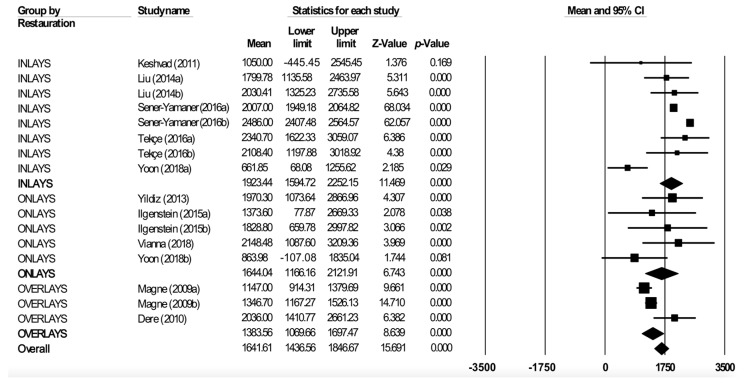
Forest plot of meta-analysis of subgroups according to restoration type (inlays, onlays, and overlays). The size of the square represents the weight of each study and the diamond represents the combined results of the meta-analysis. CI is the 95% confidence interval of the mean between a lower and an upper limit.

**Figure 6 jcm-08-01932-f006:**
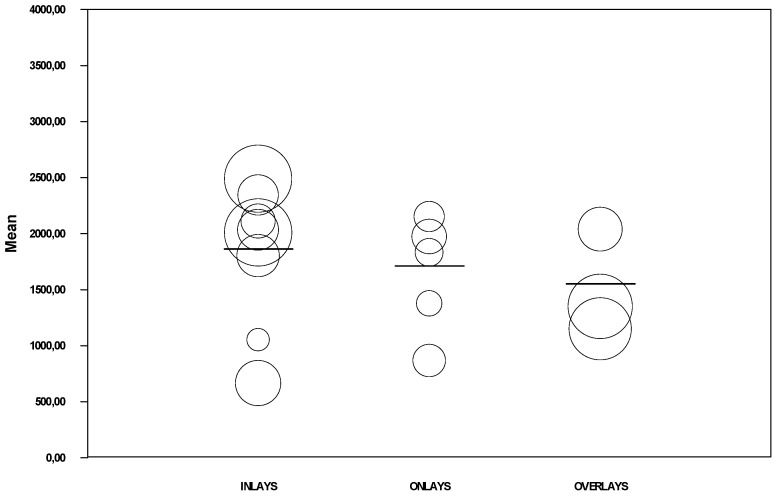
Regression model of mean fracture resistance by restoration type.

**Figure 7 jcm-08-01932-f007:**
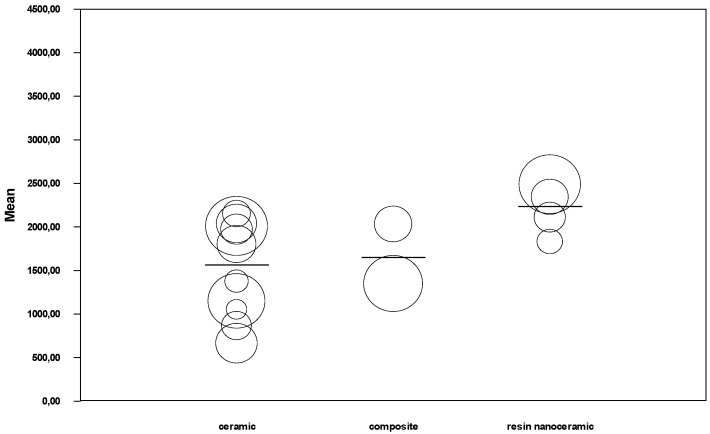
Regression model of fracture resistance by material.

**Table 1 jcm-08-01932-t001:** Main results of meta-regression model. * *p* < 0.05.

Covariate	Coefficient	95% Lower	95% Upper	Z-Value	*p*-Value
Intercept	1680.2	1324.7	2035.7	9.26	0.000
Onlays (Ref. inlays)	−151.6	−765.8	462.6	−0.48	0.628
Overlays (Ref. inlays)	−311.2	−837.4	215.0	−1.16	0.246
Composite (Ref. ceramic)	86.0	−502.5	674.6	0.29	0.774
Resin nanoceramic (Ref. ceramic)	670.9	148.5	1193.4	2.52	0.012 *

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
