# Peer review of "Fracture Resistance of Partial Indirect Restorations Made with CAD/CAM Technology. A Systematic Review and Meta-Analysis"

_jcm, 2019, doi:10.3390/jcm8111932_

Round 1
Reviewer 1 Report
Line 39: evaluate whether to modify "dentist's skill" with a more appropriate terminology such as "depending on the adopted clinical protocols"
Lines 147 and 149 are inverted
In general the article is clear and essential. Statistical analysis performed rigorously.
Can I suggest in the introduction to present the clinical indications of the materials analyzed by the selected articles and a brief description of them in order to provide the less experienced reader with a context? Also indicate the bibliographic references
Author Response
Reviewer 1.
Thank you so much for your valuable comments.
Line 39: evaluate whether to modify "dentist's skill" with a more appropriate terminology such as "depending on the adopted clinical protocols"
We find it very appropriate; therefore we have changed it in the manuscript.
Lines 147 and 149 are inverted
Again, we appreciate your review. In the manuscript´s original Word file it was fine, the mistake is due to the conversion to Pdf file. We have corrected in the article.
Can I suggest in the introduction to present the clinical indications of the materials analyzed by the selected articles and a brief description of them in order to provide the less experienced reader with a context? Also indicate the bibliographic references
We find the suggestion very interesting, and so we have added it in the introduction.
Researchers, as well as manufacturers, seek an ideal material for fabricating partial coverage restorations such as inlays, onlays, and overlays. In recent decades the numbers of ceramic or ceramic-like materials available have increased dramatically often making it difficult to define their different uses and to establish clear selection criteria and adequate indications. In 2015, Gracis et al published a classification of dental ceramics and ceramic-like materials highlighting ceramics with vitreous matrix or glass-ceramics; this category includes conventional feldspathic ceramics, synthetic ceramics (leucite-reinforced and/or lithium disilicate) and hybrid materials or ceramics with resin matrices: resin nanoceramic and resin matrix glass ceramic ( hybrid ceramic) [11 ].
Ceramics have been the subject of extensive scientific research, which has vouched for their use in this type of restoration. They offer adequate fracture resistance (160-450MPa), good survival rates, a high elasticity modulus, and they do not suffer much abrasion and wear. Composite, which has seen improvements in its composition accompanied by the introduction of CAD/CAM processing, has been presented as an alternative to ceramics that offers certain advantages such as its resilience and repairability, although its polishing capacity, resistance to wear and fracture resistance are poorer. It is claimed that hybrid materials, resin nanoceramics and resin matrix glass ceramics offer the advantages of both ceramics and composite, with an elasticity modulus similar to the natural tooth’s dentine, and characteristics that make these materials (like composite) easy to adjust, repair or modify [12].
So we have changed the references as seen in the article.
In line 229 we have added from, leucite-reinfonce ceramic IPS Empress CAD.to clarify the type of porcelain

Reviewer 2 Report
This is a very organized and well written systematic review of the fracture resistance of indirect CAD/CAM designed/fabricated restorations of different materials. This meta analysis indicated maximum fracture resistance for inlays was obtained by restorations fabricated from resin nanoceramic. For onlays, maximum fracture resistance was obtained by lithium disilicate ceramic and for overlays, the highest fracture resistance obtained by leucite-reinforced ceramic.
I understand finding articles that fit specific criteria of this study was difficult , but the number of the articles included for this study was limited. I also suggest authors make critical comments of the potential reasons for different materials fracture resistance behaviour when used in inlay, onlay and overlay preparations.
It appears one of the exclusion criteria authors selected was "in vivo" studies; nevertheless, still some in vivo studies appeared on the list and had to be discarded later.
It seems authors only selected papers that used extracted teeth as the media. Depending on the class of teeth , size , age , storage , extracted teeth may add additional variability in the results. I wonder why authors excluded standardized preparation on homogenous plastic teeth or other synthetic media for this review?
Author Response
Reviewer 2.
Thank you very much for your comments and appreciations, we find them very interesting.
English language and style are fine/minor spell check required
Sorry about that. We did the manuscript in Spanish and ordered the translation to an official native translator Mr. William James Packer, Passport No. 49624606, NIE: X-2497886-V.
This is a very organized and well written systematic review of the fracture resistance of indirect CAD/CAM designed/fabricated restorations of different materials. This meta analysis indicated maximum fracture resistance for inlays was obtained by restorations fabricated from resin nanoceramic. For onlays, maximum fracture resistance was obtained by lithium disilicate ceramic and for overlays, the highest fracture resistance obtained by leucite-reinforced ceramic.
I understand finding articles that fit specific criteria of this study was difficult , but the number of the articles included for this study was limited. I also suggest authors make critical comments of the potential reasons for different materials fracture resistance behaviour when used in inlay, onlay and overlay preparations.
It is very difficult for us to make critical comments about the possible reasons for the fracture resistance behaviour of different materials when using different types of restorations (inlay, onlay, overlay), as it is a study review of works “in vitro” made by other authors. But despite of it, we believe it is due to the different nature of the materials, their composition and resistance to bending.
It appears one of the exclusion criteria authors selected was "in vivo" studies; nevertheless, still some in vivo studies appeared on the list and had to be discarded later.
The meta-analysis and systematic review have been carried out only with the "in vitro" studies. The "in vivo" studies have only been used to write the introduction.
Despite this, we have withdrawn the articles: Felden, A.; Schmalz, G.; Federlin, M.; Hiller, K. A. Retrospective clinical investigation and survival analysis on ceramic inlays and partial ceramic crowns: results up to 7 years. Clin Oral Investig, 1982, 2,161–167 and Otto, T.; Schneider, D. Long-term clinical results of chairside Cerec CAD/CAM inlays and onlays: a case series. Int J Prosthodont, 2008, 21, 53-59. and replaced by:Gracis, S;Thompson, VP; Ferencz, JL; Silva, NR; Bonfante, EA. A new classification system for all-cermic and ceramic-like restorative materials.Int J Prosthodont, 2015, 23, 227-235 and Arnetzl,G; Arnetzl,GV. Hybrid materials offer new perspectives. Int J Comput dent, 2015, 18, 177-186.
It seems authors only selected papers that used extracted teeth as the media. Depending on the class of teeth , size , age , storage , extracted teeth may add additional variability in the results. I wonder why authors excluded standardized preparation on homogenous plastic teeth or other synthetic media for this review?
We have excluded plastic teeth or other synthetic media from the study because they have different elasticity coefficient compared to natural teeth, only with the latter we resemble to real conditions in patients mouths. On the other hand, plastic teeth do not reproduce the enamel and dentin adhesion during the luting process which is a fundamental factor to increase a lot the mechanical resistance of these materials.
